# PARP Inhibitors as Monotherapy in Daily Practice for Advanced Prostate Cancers

**DOI:** 10.3390/jcm11061734

**Published:** 2022-03-21

**Authors:** Diego Teyssonneau, Antoine Thiery-Vuillemin, Charles Dariane, Eric Barret, Jean-Baptiste Beauval, Laurent Brureau, Gilles Créhange, Gaëlle Fiard, Gaëlle Fromont, Mathieu Gauthé, Alain Ruffion, Raphaële Renard-Penna, Romain Mathieu, Paul Sargos, Morgan Rouprêt, Guillaume Ploussard, Guilhem Roubaud

**Affiliations:** 1Department of Medical Oncology, Institut Bergonié, 33000 Bordeaux, France; g.roubaud@bordeaux.unicancer.fr; 2Department of Medical Oncology, Centre Hospitalier Universitaire Besançon, 25000 Besançon, France; antoine.thieryvuillemin@oncologyfc2.onmicrosoft.com; 3Department of Urology, Hôpital Européen Georges-Pompidou, AP-HP, Paris University, 75005 Paris, France; dcharlie8@hotmail.com; 4Department of Urology, Institut Mutualiste Montsouris, 75014 Paris, France; eric.barret@imm.fr; 5Department of Urology, La Croix du Sud Hôpital, Quint Fonsegrives, 31000 Toulouse, France; jbbeauval@gmail.com (J.-B.B.); g.ploussard@gmail.com (G.P.); 6Department of Urology, CHU de Pointe-à-Pitre, University of Antilles, 97110 Pointe-à-Pitre, France; laurent.brureau@chu-guadeloupe.fr; 7Department of Urology, Grenoble Alpes University Hospital, Université Grenoble Alpes, CNRS, Grenoble INP, TIMC-IMAG, 38400 Grenoble, France; gilles.crehange@curie.fr; 8Department of Radiation Oncology, Curie Institute, 75005 Paris, France; gaellef@gmail.com; 9Department of Pathology, CHRU Tours, 37000 Tours, France; gaelle.fromont-hankard@univ-tours.fr; 10Department of Nuclear Medicine, Scintep, 38000 Grenoble, France; mathieugauthe@yahoo.fr; 11Service d’Urologie Centre Hospitalier Lyon Sud, Hospices Civils de Lyon, 69000 Lyon, France; ruffion.alain@orange.fr; 12Equipe 2, Centre d’Innovation en Cancérologie de Lyon (EA 3738 CICLY), Faculté de Médecine Lyon Sud, Université Lyon 1, 69000 Lyon, France; 13Department of Radiology, Sorbonne University, AP-HP, Radiology, Pitie-Salpetriere Hospital, 75013 Paris, France; raphaele.renardpenna@gmail.com; 14Department of Urology, University of Rennes, 35000 Rennes, France; romain.mathieu@chu-rennes.fr; 15Inserm, EHESP, Irset (Institut de Recherche en Santé, Environnement et Travail), University of Rennes, 35000 Rennes, France; 16Department of Radiotherapy, Institut Bergonié, 33000 Bordeaux, France; p.sargos@bordeaux.unicancer.fr; 17Department of Urology, Sorbonne University, GRC 5 Predictive Onco-Uro, AP-HP, Urology, Pitie-Salpetriere Hospital, 75013 Paris, France; mroupret@gmail.com

**Keywords:** PARP inhibitors, prostate cancers, homologous recombination repair, DNA repair

## Abstract

Despite recent improvements in survival, metastatic castration-resistant prostate cancers (mCRPCs) remain lethal. Alterations in genes involved in the homologous recombination repair (HRR) pathway are associated with poor prognosis. Poly-ADP-ribose polymerase (PARP) inhibitors (PARPis) have demonstrated anti-tumoral effects by synthetic lethality in patients with mCRPCs harboring HRR gene alterations, in particular *BRCA2*. While both olaparib and rucaparib have obtained government approvals for use, the selection of eligible patients as well as the prescription of these treatments within the clinical urology community are challenging. This review proposes a brief review of the rationale and outcomes of PARPi treatment, then a pragmatic vision of PARPi use in terms of prescription and the selection of patients based on molecular screening, which can involve potential genetic counseling in the case of associated germinal alterations.

## 1. Background

Despite recent survival improvements, metastatic castration-resistant prostate cancers (mCRPCs) remain lethal and represent the fifth cause of cancer death in men, worldwide [1]. While androgen-receptor signaling is involved in the majority of mCRPCs, other pathways, such as homologous recombination repair (HRR), may take part in their progression [2]. Mutations in this pathway are observed in up to 27% of mCRPCs. *BRCA2* (12–18%), *ATM* (3–6%), *CHEK2* (2–5%) and *BRCA1* (<2%) are the most common altered genes involved in HRR [2,3]. Germinal alterations are present in roughly 30 to 50% of cases [3,4]. The relative risk of prostate cancers (PCas) in germinal *BRCA2*-altered patients is 4.65-fold higher compared to non-carriers [5]. Moreover, these alterations seem to be correlated with more often aggressive, locally advanced or metastatic PCas at diagnosis [6] and their cause-specific survival seems poorer [7,8].

HRR is involved when double strand breaks (DSBs) appear. The main reparation pathway uses the serine/threonine kinase ATM, which is recruited and activates several targets, including BRCA1, through CHK2 [9]. BRCA2 and RAD51 create a complex with PALB2 after activation by BRCA1 [10,11]. Broken DNA is then restored by the use of the respective sequence from the second chromosome [12]. Thus, HRR leads to high-fidelity reparation and avoids a loss of information compared to non-homologous end joining (NHEJ), and microhomology-mediated end joining (MMEJ). Alternative pathways, such as those implicated in the Fanconi Anemia (FANC), and the BRIP1 pathway can also activate BRCA1 [13]. The role of those proteins can be divided between sensors of the DSB, such as ATM or CHK2, and effectors of the reparation, such as BRCA1/2 or PALB2.

Poly-ADP-ribose polymerase (PARP) is a family of proteins involved in single strand break (SSB) repairs [14]. PARP1 binds to damaged DNA and recruits proteins of the base excision repair or nucleotide excision repair systems through the polymerization of ADP-ribose units from nicotinamide adenine dinucleotide, also called PARylation [15,16]. After the protein recruitment, PARP1 is released from DNA. PARPs also play a role in the transcriptional regulation of various genes, including androgen receptors [17].

PARP inhibitors (PARPis) are targeted therapies inducing a catalytic inhibition or a trapping of PARP1 and 2 avoiding SSB repair. After replication, SSB is converted into DSB. This DSB can be repaired with the HRR system in efficient cells, whereas it induces genomic instability leading to cell death in the case of HRR-deficient (HRD) cells. This phenomenon is called synthetic lethality [18]. PARPis recently improved overall survival (OS) in mCRPC HRD after next-generation hormonal therapy (NHT) [19].

This manuscript reviews the timing and the methods of how PARPis can be used in daily practice and discusses the future challenges related to their prescription.

## 2. Outcomes of PARP Inhibitors Used as Monotherapy

### 2.1. Efficacy

Four PARPis are currently in late development in PCas: olaparib, rucaparib, niraparib and talazoparib (Table 1). Olaparib was the first one to show an improvement in OS [19]. The first positive signals arose from the phase II TOPARP, astudy that enrolled 50 pre-treated mCRPC patients to receive olaparib [20]. A total of 88 percent of the HRR-altered patients (14/16 patients) had a composite response defined as an objective radiological response (ORR) based on the Response Evaluation Criteria in Solid Tumors (RECIST), or a reduction of at least 50% of the PSA serum level, or a decrease in the circulating tumor cell (CTC), compared to only 6% for the HRR-efficient patients. Based on these results, 98 HRD patients with mCRPC were enrolled in TOPARP-B to receive olaparib (300 or 400 mg twice a day) [21]. The composite response rates (CRRs) were 54.3% in the 400 mg arm and 39.1% in the 300 mg cohort. A preplanned analysis subgroup showed an important ORR for *BRCA1/2* patients (11/21, 52.4%) and *PALB2* patients (2/6, 33.3%). In parallel, the phase III study PROfound was designed to compare the efficacy of olaparib to enzalutamide or abiraterone in patients with mCRPCs previously treated with at least one NHT [19,22]. All patients had an alteration (mono or biallelic) in HRR genes detected using the Foundation Medicine^®^ test on tissue from primary or metastatic sites. Prespecified HRR genes were *BRCA1, BRCA2*, *ATM*, *BRIP1*, *BARD1*, *CDK12*, *CHEK1*, *CHEK2*, *FANCL*, *PALB2*, *PPP2R2A*, *RAD51B*, *RAD51C*, *RAD51D*, and *RAD54L*. A total of 2792 (69%) samples were successfully tested over the 4047 patients screened for HRR alterations on tumor tissue. Following an analysis of these samples, 778 (28%) patients were separated between cohort A (*BRCA1/2* or *ATM* alterations) and cohort B (other HRR alterations) and were randomly assigned, in a 2:1 ratio, to receive olaparib (300 mg twice daily) or an NHT. Randomization was stratified according to previous taxane use and measurable disease. A total of 162 patients in cohort A and 94 patients in cohort B received olaparib, while 83 patients in cohort A and 48 patients in cohort B received a physician’s choice of NHT. The study was positive with an improvement of the imaging-based progression-free survival (rPFS) in cohort A (primary endpoint), which was longer in the olaparib group (7.4 months vs. 3.6 months, Hazard Ratio (HR) 0.34, 95% CI 0.25–0.47, *p* < 0.001). ORR and OS were also higher in the olaparib group of cohort A: 33% vs. 2%, and 19.1 months vs. 14.7 months, (HR 0.69, 95% CI 0.50–0.97, *p* = 0.0175) respectively, while 67% of the patients in the control arm crossed-over to receive olaparib, following radiographic progression. Exploratory analysis did not show any differences between somatic and germinal *BRCA* alterations [19]. Indeed, rPFS, ORR and OS were in the same order of magnitude between the 40 patients with germinal alterations and the 24 patients with somatic alterations in the olaparib group. Moreover, olaparib was well tolerated with manageable toxicities (mostly cytopenia, fatigue and decrease in appetite), and was associated with better health-related quality of life functioning over time, compared with the control arm [23]. Although survivals were numerically longer, no statistically significant differences on PFS or OS were observed in cohort B and the ORR results were not mentioned. In the overall population (cohorts A and B), the corresponding durations were 17.3 months and 14.0 months with a hazard ratio for death of 0.79 (95% CI, 0.61–1.03); when it was adjusted for crossover, the hazard ratio for death was 0.55 (95% CI, 0.29–1.06). Given these results, olaparib was approved by the Food and Drug Administration (FDA) for patients with an alteration in 14 of the 15 prespecified HRR genes. Indeed, the preclinical data, generated after the PROfound design, suggest that *PPP2R2A* loss of function does not confer sensitivity to PARPis, and was then withdrawn from the FDA list [22]. The European Medical Authority (EMA) was more conservative and gave its approval only for *BRCA1/2*, given the concern for a lower benefit regarding *ATM*. After the TOPARP-B results, the inclusion of *BRCA1/2*- and *ATM*-altered patients in the same cohort raised questions and gave rise to a gene-by-gene analysis that will be discussed later on in the paper [24].

Rucaparib was evaluated in the TRITON-2 phase II study. Patients with an mCRPC progressing after at least one NHT and a taxane-based chemotherapy, also presenting an alteration in the HRR genes detected by Foundation Medicine^®^ on plasma or tissue samples, received rucaparib 600 mg twice daily. For the 115 *BRCA1/2*-altered patients, the PSA response rate was 54.8% (95% CI 45.2–64.1%) and the ORR was 43.5% (95% CI 38.1–63.4%) for the 62 ORR-evaluable patients [25]. Exploratory analyses did not present any differences regarding ORR or PSA-RR between the germinal or somatic alterations. However, the PSA-RR seemed smaller in the *BRCA1* (15.4%; 2 of 13 patients) or mono-allelic (11.1%; 1 of 9 patients) groups, compared to *BRCA2* (59.8%; 61 of 102 patients) or biallelic (75.0%; 27 of 36 patients) patients, even if the populations were limited [25]. The results were mixed in the non-BRCA population. Indeed, almost no responses were observed in the *ATM, CDK12* and *CHEK2* cohorts, while encouraging outcomes were reported for the smaller groups, such as *PALB2, BRIP1,* and *RAD51B* [26]. The TRITON-2 trial generated hypotheses that suggested a better efficacy for biallelic patients, but no difference between the germinal or somatic alterations, although the populations were limited. The TRITON-3 phase III study (NCT02975934) compares rucaparib to NHT or docetaxel (physician’s choice) after 1 NHT, but no previous chemotherapy, for patients with mCRPC and a deleterious mutation of *BRCA1/2* or *ATM*.

A phase II trial, GALAHAD, assessed the efficacy of Niraparib (300 mg once a day), in patients with mCRPC with biallelic alterations in HRR genes assessed by plasma or tissue samples, and progressing after at least a taxane and one NHT. The final analysis showed a PSA response rate of 43% (95% CI, 34.7–51.5%) and an ORR of 34.2% (95% CI, 23.7–46.0%) in the *BRCA1/2* group, while in the non-*BRCA* group, the PSA response rate and ORR were 5% (95% CI, 1.4–12.2%) and 10.6% (95% CI, 3.5–23.1%), respectively [27]. The authors did not individually describe the non-*BRCA* genes. Given the small sample size, the results have to be interpreted with caution. However, as for the other PARPis, niraparib efficacy seems to be greater for *BRCA*-altered patients, and is in the same order of magnitude (ORR of 33%, 43.5% and 34.2% in cohort A of PROfound, TRITON-2 and GALAHAD, respectively), even if we expect greater efficiency in a pure biallelic population [22,25,27].

Finally, talazoparib is evaluated at the dose of 1 mg daily, in the ongoing phase II trial TALAPRO-1, in patients with mCRPC with mono or biallelic alterations in HRR genes and progressing after a taxane and one NHT. The method of assessment of the HRR status is not described. An interim analysis was recently updated [28,29]. The ORR was 43.9% (18/41 patients) for the *BRCA*-altered patients, 33.3% (1/3 patients) for *PALB2*, and 11.8% (2/17 patients) for *ATM*. No objective response was observed for the 14 other patients, and details were not presented regarding the number of patients per altered genes. To our knowledge, there is no phase III trial investigating talazoparib as monotherapy.

These trials have shown that PARPis, used as monotherapy, are efficient and constitute a treatment option for patients with often heavily pretreated mCRPC and at least *BRCA1/2* alterations. However, focusing on treatment-type sequences, the best settings in which to use a PARPi before or after docetaxel remains to be defined. The data regarding the subgroups of patients enrolled in TRITON-3, including docetaxel in the standard arm, will be highly informative.

### 2.2. Toxicity

The most common adverse events (AEs) among the patients receiving olaparib in the PROfound study were anemia, nausea, or vomiting and fatigue [19]. Around 50% of the patients experienced grade 3 or more adverse events, mostly represented by anemia (23%); other adverse events were marginal and inferior to 2 or 3%. A total of 40 percent of the patients in the control group experienced grade 3 or more toxicity, mostly represented by anemia (5%) and fatigue (5%). Olaparib was stopped in 14% of the cases due to toxicity; anemia was the most common adverse event leading to treatment discontinuation (7%), followed by cytopenia (1%) and nausea (1%) [19]. Surprisingly, pulmonary embolisms were reported in 11 patients (4%) of the olaparib arm, compared to 1 patient (1%) in the control group. This adverse event was not specifically reported in the other main studies with olaparib, and may be explained by the association with androgen deprivation therapy [30,31,32]. The tolerability of olaparib was not influenced by the burden of bone metastases or a previous treatment with docetaxel [33].

The adverse events were comparable for rucaparib, niraparib, and talazoparib. Thus, in TRITON-2, 61% of patients experienced grade 3 or more toxicity, the most frequent ones being anemia (25%) and fatigue (9%) [25]. In the interim analysis of the GALAHAD trial, the most frequent AE (all grade) was anemia (29%), thrombocytopenia (15%), and neutropenia (7%), easily managed with dose interruption or modification [27]. Finally, only the all-grade toxicity was reported in the interim results of the TALAPRO-1 study, mostly anemia (42.5%) and nausea (32.7%), comparable with PROfound and TRITON-2 studies (29) [19,25]. Of note, an increase in blood creatinine was observed in around 10–15% of the patients treated with olaparib, rucaparib, and niraparib. This issue is often resolved with treatment holds and is due to the inhibition of renal transporters (MATE-1 and MATE2-K) for olaparib and rucaparib, and probably to hemodynamic impairment for niraparib [34,35,36].

In addition to acute toxicity, some late adverse events were observed. In a recent meta-analysis, an increased risk of myelodysplastic syndrome and acute myeloid leukemia were described [37]. Based on 18 randomized controlled trials comparing PARPis to control treatment (placebo or non-placebo), the risk of myelodysplastic syndrome and acute myeloid leukemia were higher in the PARPi group compared to the control group with a Peto odds ratio of 2.63 (95%CI 1.13–6.14, *p* = 0.026) and no between-study heterogeneity (*I*^2^ = 0%, χ^2^ *p* = 0.91). The incidence was 0.73% in the PARPi group and 0.43% in the control group. In available cases, the median treatment duration was 9.8 months (3.6–17.4 months); median latency since first exposure to PARPi was 17.8 months (8.4–29.2 months). PARPis were mostly represented by olaparib (1613 patients) and veliparib (1582 patients). While in patients with prostate cancer, PARPis were not assessed following platinum salt chemotherapy and might have less risk to develop those AEs, they should be carefully followed in case of lasting hematologic toxicity.

### 2.3. How to Use in Daily Practice?

PARPis have common prescriptions rules, but they also have their own particularities. Patients should have a bi-monthly complete blood count for the first 3 months, and monthly thereafter to look for cytopenia, especially anemia. Anti-emetic drugs, such as dopamine antagonists, should be prescribed to prevent nausea. The recommended starting dose for olaparib is 300 mg twice a day (2 tablets twice a day), 600 mg twice daily (2 tablets twice a day) for rucaparib, 300 mg once a day (3 tablets) for niraparib and 1 mg daily (1 tablet) for talazoparib. The number of tablets a day can be of importance in old polymedicated patients.

Rucaparib and Niraparib do not need dose modifications in case of mild or moderate renal impairment, whereas a dose modification should be performed for olaparib and talazoparib. There is no sufficient data to prescribe PARPis in case of severe renal impairment (creatinine clearance < 30 mL/min). PARPis have not been studied in moderate and severe hepatic impairment (accessdata.fda.gov, accessed on 07 March 2022).

Finally, olaparib and rucaparib can have drug–drug interactions with inducers or inhibitors of CYP450 as they are metabolized by the enzyme, talazoparib interacts with P-glycoprotein inducers and inhibitors, while niraparib does not have major interactions. Once again, this point might be an issue in the setting of an older population with a substantial polypharmacy (accessdata.fda.gov, accessed on 07 March 2022).

## 3. Which Patients Are Candidates for Treatment with PARP Inhibitors in Daily Practice?

### 3.1. Candidate Genes

As previously discussed, good responses to PARP inhibitor treatment were observed for *BRCA1/2*-altered patients [19,20,21,25,27,29]. Efficacy seems to be in the same order of magnitude across the different PARPis. It is important to note that most studies do not differentiate between *BRCA1* and *BRCA2*. However, *BRCA1* alterations are far less frequent than *BRCA2* ones (less than 2% compared to 12–18%, respectively) [2,3]. Thus, *BRCA1* cohorts are small, for example, only 11 and 13 patients had *BRCA1* alterations in the PROfound and TRITON-2 studies, respectively, and it is difficult to assess efficacy for these patients [22,25].

On the other hand, insufficient PSA or radiological responses were reported for *ATM, CHEK2*- and *CDK12*-altered patients, irrespective of the PARPis used [19,20,21,25,27,29]. In the gene-by-gene analysis of the PROfound study, no meaningful difference of rPFS or OS was shown for *ATM* or *CDK12* [24]. This may be explained by the fact that they are sensors and not directly involved in the DNA reparation, so their action may be diluted.

*CDK12* inactivation has recently been shown to be associated with more aggressive PCs characterized by the presence of neoantigens and the enrichment of CD4 + FOXP3- tumor-invasive lymphocytes, which may be immunosuppressive [38,39]. Thus, *CDK12* alterations may be associated with a better response to immune checkpoint inhibitors rather than PARPis, as suggested by a retrospective study [40]. For this reason, *CDK12* was not considered as an HRR and was excluded from the TALAPRO-1 study.

*PALB2,* which is an effector of DNA reparation, seems promising. Indeed, the ORR was 33.3% for evaluable patients in TOPARP-B and TALAPRO-1, however the cohorts were small (2/6 patients and 1/3 patients, respectively) [21,29]. Data are missing for other candidate genes, and further explorations are needed.

Genes should be considered as predictive of responses, only if the alteration is identified as pathogenic or likely pathogenic. But even for these variants of *BRCA1/2,* the question of the zygosity remains unclear; as they are tumor suppressor genes, we would expect their function to be impaired when 2 of the alleles are altered or lost. However, none of the above-mentioned studies described the impact of the zygosity for the response, expect TRITON-2 [25]. In this last study, the PSA response rate was 11.1% (1 of 9 patients) for mono-allelic patients, compared to 75% (27 of 36 patients) for biallelic ones. Subgroups are small and warrant further exploration. Finally, the germline or somatic status of the alteration does not seem to have an impact over the response in TRITON-2 [25].

Based on these results, the FDA approved olaparib for men with an mCRPC progressing after at least one NHT, and harboring at least one altered gene among 14 of 15 genes screened in the trial (*PPP2R2A* excluded), regardless of the zygosity or the germline/somatic status. Rucaparib was also approved, but only for *BRCA1/2* patients previously treated with at least 1 NHT and 1 taxane-based chemotherapy. In Europe, the approval of olaparib was only granted for *BRCA1/2*-altered patients with mCRPC treated with a previous NHT. Other PARPis are not yet approved for prostate cancers.

### 3.2. How to Screen Patients

HRR alterations seem to occur early in PCa development and it appears that there is no enrichment with treatment selection pressure [41,42,43]. Therefore, germline and/or tumor HRR mutations should be assessed once, with a good method, and the tests should not be repeated. The European Society of Medical Oncology (ESMO) provided guidance for precision medicine, and every patient with metastatic PCa should at least be tested for somatic alterations of *BRCA1/2* [44,45]. The National Comprehensive Cancer Network (NCCN) recommends to search for somatic alterations of *BRCA1/2, ATM, PALB2, FANCA, RAD51D, CHEK2* and *PALB2* for metastatic PCas, and microsatellite instability for mCRPCs (www.nccn.org, accessed on 7 March 2022). Mutations should be sought using Next-Generation Screening (NGS), preferentially with a larger panel than those described earlier, if possible, as it is reliable, becoming cheaper, and allows for the accumulation of data for future research.

Tumor tissue testing is usually the gold standard. However, attrition rates of approximately 30–40% were observed in recent data relying on tumor biopsies, highlighting the need for improvement [3,22,46]. Failures seem to be due to material insufficiency (a small amount of tumor tissue collected during the biopsy, or material depletion for diagnosis or limited tumor content) or quality deficiency of the sample (DNA degradation during fixation or storage) [3,22]. To be more efficient, targeted tissues should be carefully selected (size of the biopsy and organ site) and collected (Figure 1).

A large tissue collection, such as a radical prostatectomy, can provide an adequate amount of material. However, if the tumor area is small, the quantity of DNA for analysis can be limited, and sometimes multiple well-driven needle biopsies can provide greater DNA amounts, although there is a risk of diluting tumoral heterogeneity.

The origin of collected samples is also important, since visceral or soft tissue should be preferred over bone metastases. Bone samples require decalcification, which may lead to DNA degradation, but the use of EDTA may limit this issue [47]. Unfortunately, bone lesions are the first site of metastases (80–90%), mostly osteoblastic and for roughly 50% of patients with mCRPCs, the only site available for biopsy [48].

Biopsy samples should be fixed using neutral formalin, as short as possible (2–6 h, maximum 24 h). Blocks with sufficient neoplastic cell content (10–30%) should be identified by the pathologist and 5 to 10 sections of 5–10 μm provided for testing [49]. Finally, the success rate is lower for old tissue samples (>10 years), which is often the case for diagnostic biopsies and prostatectomy specimens [50,51].

Considering all these limitations, liquid biopsies (NGS on plasma-derived circulating-tumor DNA (ctDNA)) are being developed. Indeed, they are easy to obtain, fast to assess, cost the same as tissue biopsies, and they are reliable. Analysis from the PROfound trial showed that 81% (503/619) of ctDNA samples tested yielded a result [52]. The reported positive and negative percentage agreements were 81% (95% CI 75–87%) and 92% (95% CI 89–95%), respectively, for the *BRCA* and *ATM* status between tissue and ctDNA samples, with tissue as reference [52]. Baseline characteristics and proportions of *BRCA1/2* and *ATM* between tissue and ctDNA-positive patients in cohort A were almost the same. rPFS improvement was consistent for *BRCA* and *ATM*-altered patients on ctDNA (HR 0.33, 95% CI 0.21–0.53) and in the same order of magnitude as the improvement observed in the Intention to Treat (ITT) population identified by tissue (HR 0.34, 95% CI 0.25–0.47) [53]. Moreover, ctDNA can provide information about clonal heterogeneity. However, not all patients have concordant results, the allelic fraction is not easy to estimate using liquid biopsies, and clonal hematopoiesis of indeterminate potential (CHIP) can interfere with results. CHIP corresponds to an age-related phenomenon where somatic alterations are observed in a clonally expanded subpopulations of blood cells. CHIP can be present in 10 to 20% of PC patients; they often concern HRR genes and make the ctDNA interpretation difficult in case of low tumor fraction [54]. The timing of blood harvest is key in prostate and should not be performed while the patient is responding because of a low yield of ctDNA [55].

When possible, combining both tissue and liquid biopsies may be valuable and may lower the attrition rates and limits of either method.

### 3.3. When Should Homologous Recombination Repair Testing Be Performed?

As previously observed, HRR alterations seem to occur early in PCa development with marginal enrichment under treatment selection pressure [41,42,43]. Therefore, HRR status could be determined at any time of the medical care. However, ctDNA analysis should be performed in case of disease progression, before the initiation of a new treatment line, as ctDNA concentration rapidly decreases in the case of effective treatment [55].

An early determination of the HRR status can help the physician to choose the best treatment option, as HRR-altered PCa seem to be more aggressive and correlated with a shorter cause-specific survival [6,7,8]. Currently, PARPi prescription should follow the PROfound design and be used after at least one NHT, regardless of previous chemotherapy exposure [22]. Contrary to ovarian and pancreatic cancers, the treatment indication does not follow platinum sensitivity. However, further research is needed to define the role of platinum, and its use after resistance to PARPis.

In summary, a somatic determination of HRR status should be conducted for all patients with mCRPC, in order to administer PARPis after progression under NHT. Somatic determination should be considered even for castration-sensitive prostate cancer (mCSPC) to include patients in clinical trials. Germinal HRR alterations should be looked for in cases of family history, even for non-metastatic PCa, to perform individual screening for relatives and discuss clinical trials for the patient. It is important to stress that germinal screening cannot replace somatic screening for mCRPCs, since roughly half of the altered patients carry only a somatic mutation [3].

### 3.4. Consequence of Germinal Mutation with Genetic Counseling

Three different specific cases were described: metastatic PCa, non-metastatic PCa, and men without PCa (Figure 2). For the germinal testing of *BRCA2, BRCA1* and Mismatch Repair (MMR) genes, the Philadelphia PC Consensus Conference, NCCN and European Association of Urology (EAU) recommend to address all metastatic PCs [56,57,58]. However, as discussed earlier, all men with PC should have tumor testing with a broad gene panel. It is easier for the geneticists to provide consultative care only to patients with alterations found in tumor DNA. Patients should be informed of the possible implications of having an alteration on genes involved in DNA repair, because they will also be subsequently tested for germinal alterations.

Genetic counseling is recommended for patients with non-metastatic PCs and a family history of PCs (depending on the recommendations, family history criteria differ) [56,57,58]. It should be considered for aggressive diseases (≥T3a, intraductal, ≥Gleason 8, positive lymph nodes), Ashkenazi Jewish Ancestry, ≥2 relatives with cancer in the spectrum of hereditary breast–ovarian cancer or Lynch syndrome. Finally, men without PCs should be addressed to geneticists in case of a family history of PCs and consultations should be considered if they have 2 or more relatives with hereditary breast–ovarian cancer or cancer of Lynch spectrum.

## 4. Perspectives

While sustained efficacy is observed with PARPis alone, primary and secondary resistances are reported, and different ways are explored to overcome them. Several studies have shown that the inhibition of the androgen-receptor pathway downregulates the transcriptional program of DNA repair genes, conferring a BRCAness state [59,60,61]. Moreover, PARP-1 promotes PC cell growth and progression through AR [17]. To study the potential synergistic action, Clarke et al. randomized 142 patients with mCRPCs previously treated with docetaxel, but not any NHT, regardless of HRR status, to receive abiraterone with olaparib or placebo [62]. Only 21 patients (15%) had HRR alterations and 35 patients (25%) were considered as wild types. The 86 (61%) remaining patients were unclassified because of invalid plasma, tissue or germline tests. At the data cut-off, rPFS was improved in the experimental group (13.8 months vs. 8.2 months, *p* = 0.034), but no significant difference was observed in the prespecified analysis for HRR-altered and HRR wild-type patients, probably because of the size of the populations. The results for OS were immature. A second analysis of the germline and ctDNA samples assessed the HRR status more accurately, increasing the HRR characterized population to 68% of the 142 randomized patients, with a high concordance between the methodologies. The rPFS was significantly increased in the experimental arm for the 73 HRR wild-type patients (HR 0.54, 95% CI: 0.32–0.93), and a trend was observed for the 23 HRR-altered patients (HR 0.62, 95% CI: 0.23–1.65), probably due to the size of the population [63]. Of note, the toxicity was higher in the combination arm compared to the control arm (54% vs. 28% of grade 3 or more AE, respectively), especially in terms of serious cardiovascular events (7 vs. 1). These adverse events led to death in almost 3% of the cases. While such a combination is promising, it should be used carefully in patients with cardiovascular history. Based on these results, the PROPEL trial randomized abiraterone vs. abiraterone and olaparib as first line for mCRPCs (all comers). The study met its primary endpoint with an rPFS improvement in the combination arm (24.8 months vs. 16.6 months, *p* < 0.0001) (Saad et al. ASCO-GU 2022). In a subgroup analysis, efficacy seemed to be observed regardless of HRR status (HR 0.62 and HR 0.54 for HRD and HRR-efficient patients, respectively). Contrary to the study of Clarke et al., serious cardiovascular events were balanced between the groups. The results of the MAGNITUDE trial were also released at ASCO-GU 2022. It randomized abiraterone vs. abiraterone and niraparib as the first line for mCRPCs, and the patients were divided regarding their HRR status (Chi et al. ASCO-GU 2022). The HRR-efficient cohort was prematurely closed for futility. An rPFS improvement was observed in the experimental arm of the HRD cohort (16.5 months vs. 13.7 months, *p* = 0.02). Serious cardiovascular events were similar in the 2 arms. This combination seems promising and could confer efficacy to PARPis, even in the HRR-efficient population. However, the results are opposed in HRR-efficient cohorts in PROPEL and MAGNITUDE trials, and no explanation has been found for this difference. Other ongoing studies are recruiting to address the question of the combination of NHT and PARPis for mCRPC or mCSPC better [18].

The efficacy of immunotherapy for prostate cancer remains unclear. After promising results with OS improvement using sipuleucel-t, the outcomes of check-point inhibitors are mixed [64,65,66,67]. PARPis can act as immunomodulatory agents, inducing PD-L1 expression through interferon (INF) after the activation of the cGAS/STING pathway via the accumulation of unrepaired DNA fragments [68]. In the KEYNOTE-199 study, 2 of the 6 patients responding to pembrolizumab, for mCRPC refractory to docetaxel, had an HRR gene alteration [67]. The largest published trial to explore the combination of immunotherapy and PARPis enrolled 41 patients with heavily pretreated mCRPCs to receive pembrolizumab and olaparib, regardless their HRR status [69]. None of the patients had HRR gene alteration. The CRR was low, around 10%, comparable to the patients without HRR alteration of the TOPARP-A study (9%), suggesting no additional effects of immunotherapy [20]. However, even if a synergy was not observed for the CRR, pembrolizumab may prolong PARPi efficacy and longer follow-up may confirm this hypothesis.

Larger phase III studies, such as the KEYLINK-010 study (NCT03834519) comparing pembrolizumab with olaparib to an NHT for previously treated mCRPC, irrespective of their HRR status, will help us explore the combination of PARPis with immunotherapy. Other combinations with chemotherapies, such as platinum, alkylating agents or topoisomerase inhibitors, or other DNA damage-response inhibitors, such as ataxia telangiectasia and rad3-related kinase (ATR) inhibitors, are under investigation [70,71,72,73]. PARPis are also studied in combination, at earlier stages, such as localized prostate cancers or metastatic castration-sensitive prostate cancers.

## 5. Conclusions

Treatment with PARPis significantly improves overall survival for mCRPC patients. Roughly one in four patients harbor a somatic or germinal alteration of HRR genes. Similar to ovarian cancers, *BRCA* alterations seem to be the most reliable biomarker to predict PARPi efficacy [30]. Further studies are needed to address the implication of other HRR genes’ response to PARPis. Somatic HRR alterations should be sought for all patients with metastatic PCas, using, where possible, fresh visceral tissue and/or liquid biopsies. Screening for germinal *BRCA* mutations and genetic counseling must be organized in cases of family history or somatic alterations. Current recommendations indicate that PARPis should be used alone, after at least one NHT, but recent results of trials investigating combinations with NHT raise questions in order to delineate their use better, in sequence or in combo.

## Figures and Tables

**Figure 1 jcm-11-01734-f001:**
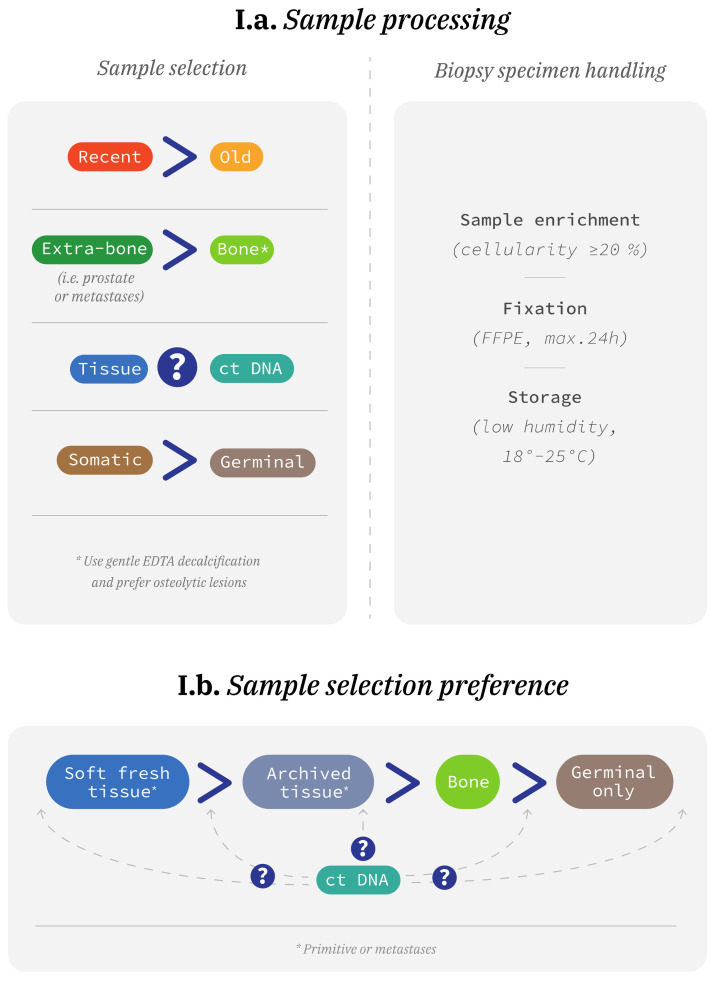
Modality of screening for homologous recombination repair deficiency. (**a**) To improve screening quality, pathologists should favor recent tissue samples, from extra-bone localizations. Germinal screening should be considered only if there is no tumor DNA available. Pathologists should select a tumor area with good cellularity, use a gentle fixation process, and store samples in controlled conditions. (**b**) Where possible, pathologists should use soft fresh tissue rather than archived tissue or bone lesion to determine homologous recombination repair status. Germinal analysis can be considered only in the absence of tumor DNA. Analysis of circulating tumor DNA (ctDNA) is complementary and its role and place in the sequence is yet to be defined.

**Figure 2 jcm-11-01734-f002:**
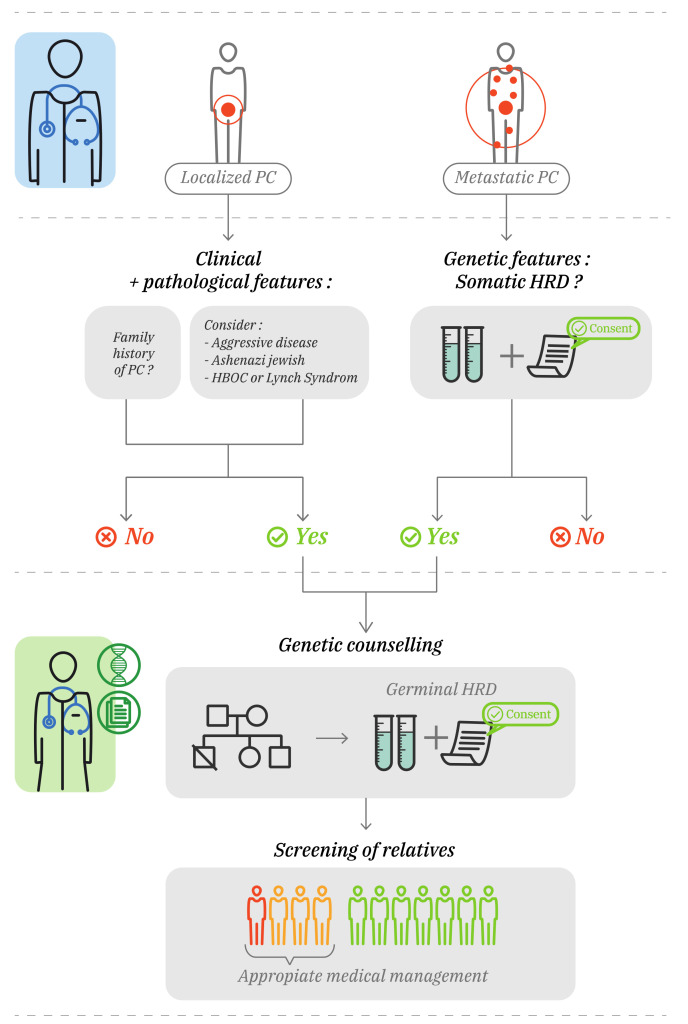
Genetic counselling place for prostate cancer. Localized prostate cancer (PCa) patients should be addressed for genetic counselling in cases of family history of PC and the consultation can be considered for aggressive disease ((≥T3a, intraductal, ≥Gleason 8, positive lymph nodes), Ashkenazi Jewish and relatives with cancers in the spectrum of Lynch or HBOC syndromes. Every patient with metastatic PC should have tumor determination of homologous recombination repair (HRR) status. Patients with HRR deficiency (HRD) found in the tumor should be addressed for genetic counselling to look for a germinal HRD. Legend: Red man: carrier of PCa and HRD. Orange men: healthy carriers of HRD needing an appropriate medical management. Green men: non carriers of PCa or HRD.

**Table 1 jcm-11-01734-t001:** Phase II or III trials using PARP inhibitors alone to treat prostate cancers with results. *ATM*: ataxia telangiectasia mutated, s*BRCA*: somatic deleterious mutation of *BRCA*. g*BRCA*: germinal deleterious mutation of *BRCA*. mCRPC: metastatic castration-resistant prostate cancer, NHT: new hormonal therapy. HRR: homologous recombination repair, HRD: homologous repair deficiency, CTID: clinical trial identification, OS: overall survival, PFS: progression free survival, ORR: objective response rate, PSA response rate: decline of more than 50%. ◊Germline or somatic alteration specified. ⊕Composite response rate: response according to RECIST or PSA reduction >50%, or reduction of circulating tumor cells to less than 5/7.5 mL of blood confirmed 4 weeks later.

CTID	Treatment	Phase	N Patients or Estimated Enrollment	Disease Status	Mandatory HRR Status for Inclusion	Determination Method for HRD	Primary Endpoints	Results
**NCT01682772/ TOPARP-A**	Olaparib	2	50	mCRPC after at least docetaxel	No	Tumor	⊕Composite response rate	All comers: 33% HRD: 88%
**NCT01682772/ TOPARP-B**	Olaparib	2	98	mCRPC after at least docetaxel	Bi-allelic deleterious HRD	Tumor	⊕Composite response ratePreplanned secondary endpoint: ORR	*BRCA1/2*: 83%, ORR: 52.4% *PALB2*: 57%, ORR: 33.3% *ATM*: 37%, ORR: 8.3% *CDK12*: 25%, ORR: 0%
**NCT02987543/ PROfound**	Olaparib vs. NHT	3	778	mCRPC after at least 1 NHT	Bi or mono-allelic somatic or germline deleterious HRD	Tumor	Radiographic PFS Preplanned secondary endpoint: OS	rPFS: *BRCA/ATM*: 7.4mo vs. 3.6mo, HR = 0.34 (95% CI 0.25–0.47) General HRD: 5.8mo vs. 3.5mo, HR = 0.49 (95% CI 0.38–0.63) OS: *BRCA/ATM*: 19.1mo vs. 14.7mo HR = 0.69 (95% CI 0.5–0.97) No-*BRCA/ATM*: 14.1mo vs. 11.5mo HR = 0.96 (95% CI 0.63–1.49)
**NCT02854436/ GALAHAD**	Niraparib	2	291	mCRPC after at least 1 chemotherapy and 1 NHT	◊Bi-allelic HRD or germline pathogenic *BRCA1/2* alteration	Tumor or plasma	ORR	*BRCA*: 41% Non-*BRCA*: 9%
**NCT02952534/ TRITON-2**	Rucaparib	2	193	mCRPC after at least 1 chemotherapy and 1 NHT	Bi or mono-allelic somatic or germline deleterious HRD	Tumor or plasma	ORR and PSA response rate (PRR)	s*BRCA1/2*: 43.9%, PRR: 50.7% g*BRCA1/2*: 42.9%, PRR: 61.4% *ATM*: 10.5%, PRR: 4.1% *CDK12*: 0%, PRR: 6.7% *CHEK12:* 11.1%, PRR: 16.7%
**NCT03148795/ TALAPRO-1**	Talazoparib	2	100	mCRPC after at least 1 chemotherapy and 1 NHT	Mono or bi-allelic HRD (*CDK12* excluded)	Tumor	ORR	*BRCA*: 43.9% *PALB2*: 33.3% *ATM*: 11.8% Other HRD: 0%

## Data Availability

Not applicable.

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
