# Peer review of "PARP Inhibitors as Monotherapy in Daily Practice for Advanced Prostate Cancers"

_jcm, 2022, doi:10.3390/jcm11061734_

Round 1

Reviewer 1 Report

Teyssonneau and colleagues wrote a review about PARP inhibitors as a therapy for prostate cancers.The manuscript describes both the efficacy and the toxicity of PARP inhibitors, moreover how to use them and which patients can receive this treatment. The work is really interesting and suitable for publication after few modifications. Unfortunately figure 1 and figure 2 were missing from the manuscript and supplementary data. May the authors provide them?  I would suggest the authors to make an evaluation of the different inhibitors and which ones are the best treatment for prostate cancer. A figure resuming all the data regarding the studies they are involved in might help. 

Author Response

Updated comments:

“I would like to make a comment regarding the manuscript that you have sent me. I suggest that the figures should be mentioned in the text at the point where the topic is introduced for the first time and not in the title of the paragraph.”

We agree with reviewer #1 and mentioned figures and table in the text

Reviewer 2 Report

It is an interesting paper, reviewing the main points in the use of PARP inhibitors in clinical practice of prostate cancer. The figures are understandable and illustrative for readers.

  1. The figures must be referenced in the text and not in the subheads.
  2. Some revision in English writing to improve the quality of article. 

Author Response

Updated report:

“It is an interesting paper, reviewing the main points in the use of PARP inhibitors in clinical practice of prostate cancer. The figures are understandable and illustrative for readers.

  1. The figures must be referenced in the text and not in the subheads.
  2. Some revision in English writing to improve the quality of article.”

We agree with reviewer #2 : we mentioned figures and table in the text and made revisions with a native English-speaking medical writter

Round 2

Reviewer 2 Report

The article can be accepted for publication.